# Balance recovery schemes following mediolateral gyroscopic moment perturbations during walking

Omid Mohseni[1,2]*, Asghar Mahmoudi[1,3], Vahid Firouzi[1,4], Andre Seyfarth[1], Heike Vallery[5,6], Maziar A. Sharbafi[1]

1 Lauflabor Locomotion Laboratory, Institute of Sport Science, Centre for Cognitive Science, Technische Universität Darmstadt, Hessen, Germany, 2 Measurement and Sensor Technology Group, Department of Electrical Engineering and Information Technology, Technische Universität Darmstadt, Hessen, Germany, 3 Institute for Mechatronic Systems, Faculty of Mechanical Engineering, Technische Universität Darmstadt, Hessen, Germany, 4 Simulation, Systems Optimization and Robotics Group, Department of Computer Science, Technische Universität Darmstadt, Hessen, Germany, 5 Delft Biorobotics Lab, Faculty of Mechanical, Maritime and Materials Engineering, Delft University of Technology, Delft, Netherlands, 6 Institute of Automatic Control, Faculty of Mechanical Engineering, Rhine-Westphalia Technical University of Aachen, Aachen, North Rhine-Westphalia, Germany

* omid.mohseni@tu-darmstadt.de

**Data Availability Statement:** The data used for this study is published on TUdatalib Repository of TU

## Abstract

Maintaining balance during human walking hinges on the exquisite orchestration of whole-body angular momentum (WBAM). This study delves into the regulation of WBAM during gait by examining balance strategies in response to upper-body moment perturbations in the frontal plane. A portable Angular Momentum Perturbator (AMP) was utilized in this work, capable of generating perturbation torques on the upper body while minimizing the impact on the center of mass (CoM) excursions. Ten participants underwent upper-body perturbations during either the mid-stance or touch-down moment in both ipsilateral and contralateral directions in the frontal plane. Our findings emphasize the predominant role of the hip strategy and foot placement as primary mechanisms for recovering from WBAM perturbations, regardless of the perturbation's timing or direction. Specifically, hip add/abduction torque and step width were significantly modulated following perturbations during the stance and swing phases, respectively, to reject frontal-plane balance threats. The knee and ankle torque modulation were not found to be effective in the recovery process. Additionally, we observed that recovery from WBAM perturbations occurs promptly within the same stride in which the perturbation occurs, unlike other perturbation scenarios, such as platform translation. These insights have the potential to enhance the development of assistive devices and more robust controllers for bipedal robots.

## Introduction

The intricate coordination of the whole-body angular momentum (WBAM) plays an essential role in maintaining balance for humans [1] and humanoid robots [2, 3]. This complex task is

Darmstadt and can be found via the following DOI: https://doi.org/10.48328/tudatalib-1578.

**Funding:** This work was supported by the German Research Foundation (www.dfg.de) within RTG 2761 LokoAssist under grant no. 450821862 and the Hessian Ministry of Higher Education, Science, Research and Art and its LOEWE research priority program under the grant 'WhiteBox', both awarded to AS. The funders had no role in the study design, data collection, data analysis and preparation of the manuscript.

**Competing interests:** The authors have declared that no competing interests exist.

achieved through segment-to-segment cancellation of the angular momentum and counteracting the contributions of the ground reaction forces (GRFs) [4]. Effective regulation of WBAM through control of joint moments is indispensable for establishing a stable and efficient gait, especially for the elderly and post-stroke patients [5, 6]. When confronted with external perturbations such as tripping, slipping, or push/pull incidents that alter WBAM, an active response is necessary to prevent falls. Notably, the regulation of WBAM could take precedence in recovery strategies, even surpassing the consideration of whole-body linear momentum (WBLM) [7, 8]. Consequently, the inability to regulate WBAM increases the risk of falling, especially in the presence of perturbations. Therefore, gaining a deeper understanding of how able-bodied individuals recover from WBAM perturbations is imperative. This knowledge can be applied in designing effective rehabilitation strategies, developing assistive technologies, and successfully controlling bioinspired bipedal robots.

To further understand the strategies employed in countering WBAM perturbations, several studies applied diverse perturbations during both standing [9, 10] and walking scenarios [8, 11–13]. Considerable attention has been dedicated to investigating *foot placement* as a primary mechanism for stabilizing gait in both the anterior-posterior (AP) and mediolateral (ML) directions [14, 15]. The *ankle strategy*, involving the application of active muscle moments around the ankle of the stance foot, has also been studied extensively as another stabilization mechanism during walking [13, 16]. A further mechanism, known as the *hip strategy*, changes the angular momentum of the segments around the center of mass to alter the direction of the horizontal component of the GRF [13]. Nevertheless, it still remains unclear how recovery strategies are utilized during walking when the angular momentum in the frontal plane is perturbed.

Maintaining balance during walking in the absence or presence of perturbations becomes more challenging in ML than AP direction [17–21]. While AP perturbations are demonstrated to be rejected via passive dynamics of the body, active motor control is required for managing the perturbations occurring in the ML direction [17, 21]. Frontal-plane perturbations can result in higher rates of center of pressure (CoP) changes [22] or an increase in the GRF [23], potentially leading to greater postural misalignments [23, 24]. Consequently, active control mechanisms are essential in mitigating these challenges [25]. For example, foot placement, which is an important balance strategy amidst perturbations [26–30], plays a more substantial role in frontal plane balance maintenance as opposed to the sagittal plane [31]. Given the importance of frontal-plane balance during walking, further investigation is necessary to explore the control strategies employed, especially in response to novel perturbation scenarios.

Prior research has highlighted the complex strategies individuals use to maintain balance in a variety of destabilizing scenarios [31–34]. Most studies, however, have concentrated on perturbations caused by slipping or tripping [11, 12], treadmill accelerations and decelerations [35], external forces applied to the pelvis or shoulder via pushes or pulls [36–38], or swing leg obstacle collisions [39]. These perturbations affect both linear and angular momentum, requiring complex corrective actions across multiple response mechanisms, which complicates the isolation of specific strategies for regulating WBAM. In contrast, whole-body pitch angle perturbations, achieved by applying two simultaneous perturbations of equal magnitude in opposite directions [40], allow for angular momentum modulation without affecting linear momentum. However, such devices are typically not portable and require mechanical linkages between the body and external structures, which impose physical constraints and could limit the movement of biological joints. Additionally, these devices are restricted to perturbations in the sagittal plane, necessitating substantial mechanical modifications for perturbations in other planes.

This study employs an Angular Momentum Perturbator (AMP) integrated into a backpack, housing a flywheel that generates gyroscopic torques. These torques are designed to induce perturbations to the upper body, specifically in the frontal plane, while minimizing effects on Center of Mass (CoM) excursions and, consequently, WBLM [10, 41, 42]. This design allows the investigation of strategies employed specifically in the regulation of angular momentum during walking. Utilizing this portable system allows for a more natural evaluation of overground walking, surpassing the limitations of motorized treadmill setups and traditional force-based and whole-body pitch angle perturbations [8, 40, 43, 44]. To the best of our knowledge, this study is unique in its investigation of angular momentum perturbations in the frontal plane during overground walking with minimal effect on the WBLM, an area that has not been explored before. In this work, perturbations are randomly activated at two distinct instances of the gait cycle (right leg mid-stance and left leg touch-down), and in both ipsilateral and contralateral directions. The primary objective of this investigation is to gain insights into the underlying balance strategies by analyzing kinematics and joint moments in response to mediolateral upper-body moment perturbations during walking.

## Materials and methods

### Participants

The study involved ten able-bodied adult participants (nine male, one female) of age $34.1 \pm 14.2$ years, weight $76.1 \pm 12.4$kg, height $1.79 \pm 0.08$m (mean ± std). All subjects volunteered to participate in the research from mid-June to the end of September 2018 and provided written informed consent. The experimental protocol received approval from the Human Research Ethics Committee of Delft University of Technology (Project ID: 350), and all procedures adhered to the relevant guidelines and regulations outlined in the Declaration of Helsinki. All subjects have the same limb preference and step with the same leg onto the first force plate.

### Perturbation device

An AMP consisting of a control moment gyroscope (CMG) housed in a backpack-like structure was used as the perturbation device [10, 42]. This AMP can produce effective pure moments on the body, devoid of net translational components, by rapidly changing the orientation of the spinning wheel about an orthogonal gimbal axis [41]. This portable perturbation device departs from conventional force-based push or pull devices and allows for overground assessment.

Our perturbation device has a weight of 16kg and produces perturbation torques with a symmetric trapezoidal profile. It features a peak torque of approximately 50Nm and follows rise, hold, and fall times, each lasting 100ms [42]. This profile accounts for the gimbal motor's limited ability to accelerate or decelerate the gimbal structure and generate the desired gyroscopic effect [10]. In this study, a gimbal motor torque of approximately 12Nm was required to produce the desired perturbations.

### Experimental protocol

Participants initiated walking from a standing position, starting with their left leg. They walked three steps along a raised platform before making contact with the first and second force plates on the fourth and fifth steps (right and left steps, respectively). The walking sequence continued for an additional three to four steps before coming to a stop.

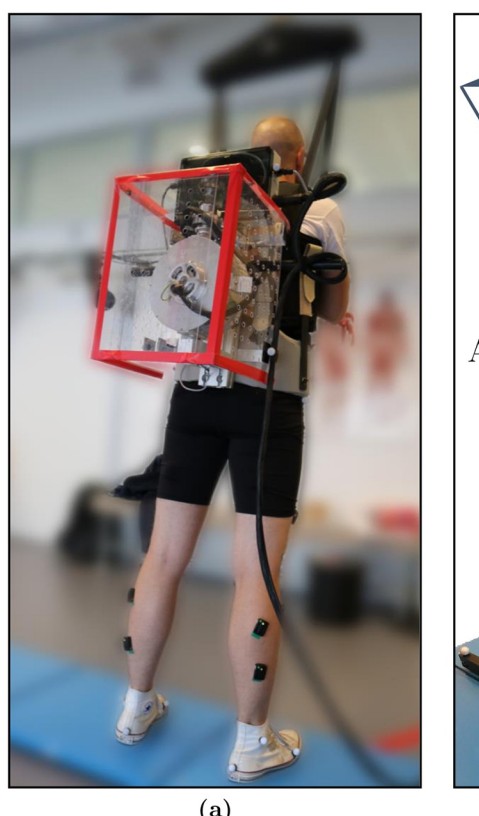
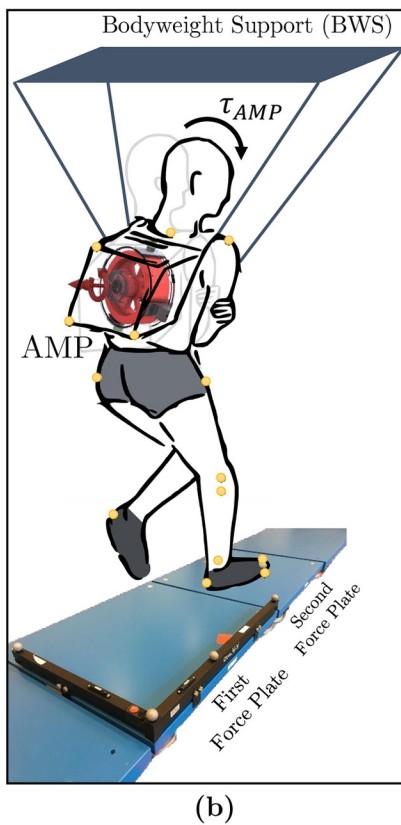
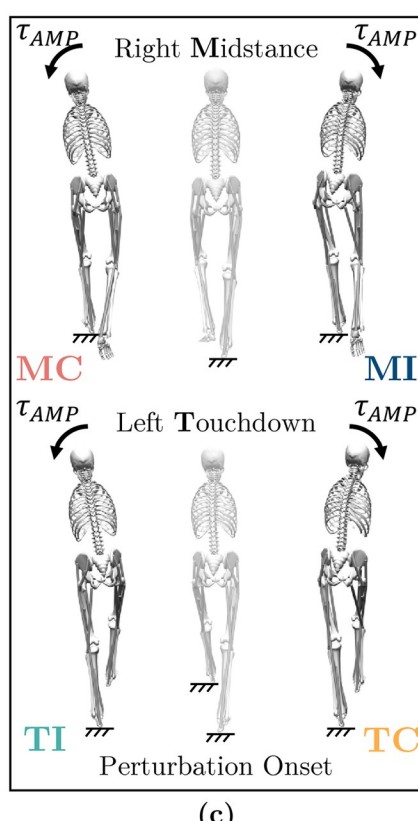

**Fig 1. Experimental setup and perturbation protocol.** (a) Participant equipped with the portable angular momentum perturbator (AMP), capable of applying pure moments to the upper body. (b) Participants initiated walking with their left leg, walking a raised platform for three steps before encountering the first and second force plates, where perturbations were applied. An additional 3 to 4 steps were taken before stopping. For safety, participants were secured with a safety harness. (c) Participants experienced four different perturbations: *MI* (Midstance-Ipsilateral) and *MC* (Midstance-Contralateral) at the midstance of the right leg on the first force plate, and *TC* (Touchdown-Contralateral) and *TI* (Touchdown-Ipsilateral) at the touchdown of the left leg on the second force plate. Stick figures visually represent each perturbation experimental condition, with the middle figure illustrating the application of perturbation and the side figures depicting the effect of perturbations.

A total of 48 trials were conducted for each participant, comprising 40 randomized trials involving active torque perturbations from the AMP and 8 control trials (*CTRL*) without perturbations. The 40 perturbation trials encompassed four conditions, each repeated 10 times, with a consistent intensity level across all participants. These conditions included contralateral and ipsilateral torque directions in the frontal plane at either the midstance of the right leg or the touchdown of the left leg (Fig 1). The four perturbation cases are referred to as Midstance-Ipsilateral (*MI*), Midstance-Contralateral (*MC*), Touchdown-Contralateral (*TC*), and Touchdown-Ipsilateral (*TI*). AMP perturbations occurred either in the middle of the fourth step or the beginning of the fifth step, respectively. To mitigate the risk of fall-related injuries during perturbations, participants wore a safety harness connected to the Rysen body weight support system (Motekforce Link, Amsterdam, The Netherlands). This system actively detects and halts the subject's falling movements [45]. To prevent vertical unloading forces during measurements, the Rysen system was set at its lowest assistance level.

## Data collection & processing

We conducted a data collection, capturing electromyography (EMG), kinematic, and kinetic data from the participants. All measurement devices, including the AMP data logging, were

synchronized through a manual trigger signal. Kinematic data were acquired using a Qualisys motion capture system (Gothenburg, Sweden) at a frequency of 200Hz. Nineteen reflective markers were placed on specific anatomical landmarks of the body, with an additional four markers on the stationary frame of the AMP.

For kinetic data, two force plates with built-in amplifiers (9260AA6, Kistler Holding AG, Winterthur, CH) positioned in the middle of the walkway (corresponding to the third or fourth steps) were employed. Data acquisition units (5695B, Kistler Holding AG, Winterthur, CH) recorded Ground Reaction Forces (GRF) of each leg at a frequency of 1000Hz. Real-time anti-aliasing low-pass filtering using a 3rd-order analog Butterworth filter with a cut-off frequency of 500Hz was applied to the GRF data. To analyze kinematics and kinetics, we utilized OpenSim 4.3, employing a 23-degree-of-freedom (DoF) full-body model with 92 musculotendon actuators representing 76 muscles for each participant. The gyroscopic backpack was integrated into the model using the four AMP markers and the calculated AMP inertia.

## Outcome measures

**Joint angles and torques.** To calculate joint angles, we conducted inverse kinematics in OpenSim using measurements from reflective markers captured by motion capture cameras. Subsequently, inverse dynamics analysis was carried out, incorporating GRF as external forces, to determine joint torques.

**Whole-body angular momentum (WBAM).** The WBAM ($H$) vector about the body's CoM position was calculated in the three dimensions from the following equation:

$$H = \sum_{i=1}^{n} [(r_i^{CoM} - r_{body}^{CoM}) \times m_i v_i^{CoM} + I_i \omega_i] \tag{1}$$

where $n$ is the number of segments (n = 12); $r_i^{CoM}$ and $r_{body}^{CoM}$ are the position vectors of the CoM of the i-th body segment and the whole-body CoM (the weighted sum of each body segment's CoM) in the laboratory frame, respectively; $v_i^{CoM}$ is the velocity vector of the i-th body segment; $m_i$ and $I_i$ are the mass and inertia tensor of the i-th body segment at the CoM, respectively; and $\omega_i$ is the angular velocity vector of the i-th body segment about its CoM.

**Step width.** The step width is determined by measuring the mediolateral distance between the calcaneus markers affixed to each foot at each heel strike during gait.

To decrease between-subject variability due to their anthropometric characteristics, all aforementioned measures were individually scaled based on the participant's height $l$ and/or mass $m$ (depending on the measure), rendering the values dimensionless [46]. Joint torque was normalized by $m.g.l$ (where $g$ = 9.81m/s$^2$), WBAM was normalized by the product of subject mass, height, and $\sqrt{g.l}$ [47, 48], and step width was normalized by height. To restore the values to their original order of magnitude, the scaled measures were multiplied by measure-specific scaling factors derived from the average height and/or mass across all participants [40].

## Statistics

For the statistical analysis, we initially assessed the normal distribution of the data using the Shapiro-Wilk test. If the data exhibited a normal distribution, subsequent tests were conducted to evaluate the homogeneity of variance (Bartlett's test) and sphericity of variance (Mauchly's test). To examine the differences between perturbed and unperturbed conditions, Statistical Parametric Mapping—SPM for one-dimensional signals [13, 49, 50] was employed. The paired t-test was utilized to compare trunk angle, joint moments, and CoM velocity against their

corresponding values from the control trials. Statistical significance was considered for p-values less than 0.05.

## Results

In the following, we initially assess the performance of AMP under various perturbation scenarios, specifically examining its impact on WBAM and upper-body kinematics in terms of trunk angle changes. Subsequently, we investigate the alterations in lower-limb joint moments in response to ML perturbations. Finally, we analyze the changes in CoM velocity following the perturbation and explore its correlation with step width.

### Perturbation profile and effect

The AMP perturbations were designed to adopt a symmetric trapezoidal profile. However, due to the motor dynamics and the influence of the low-level feedback loop controlling the gimbal motor's reference torque [41], the actual output torque exhibited a curved trapezoidal shape. As depicted in Fig 2, the ML moment profiles generated by AMP reached peaks ranging between + 45.3Nm to + 50.0Nm for rightward (*MI* and *TC*) perturbations, and peaks ranging from −48.7Nm to −52.1Nm for leftward (*MC* and *TI*) perturbations. The peak torque generated by AMP occurred approximately between 170ms and 270ms after the onset of the perturbation. Consequently, the trunk angle, WBAM, and the time derivative of WBAM exhibited statistically significant alterations with respect to control trials in response to these torque profiles.

Due to the perturbations induced, the trunk angle displayed ML deflection peaks of + 6.9deg to + 10.0deg for rightward (*MI* and *TC*) perturbations, and peaks ranging from −11.7deg to −13.4deg for leftward (*MC* and *TI*) perturbations, relative to the trunk posture observed in control trials (*CTRL*) without perturbations.

Regarding whole-body angular momentum (WBAM) in the frontal plane, the AMP perturbations significantly elevated WBAM peaks in alignment with the direction of perturbation. For rightward perturbations (*MI* and *TC*), WBAM peaks increased to between + 0.11 and + 0.15. Conversely, leftward perturbations (*MC* and *TI*) led to initial WBAM peaks ranging from −0.11 to −0.12.

The net ML torque, derived as the first derivative of whole-body angular momentum, revealed that immediately after the perturbation, the net torque spiked due to the sudden change in angular momentum caused by the external torque. For rightward perturbations, $\dot{H}$ initially peaked up to + 0.73s$^{-1}$ and + 1.07s$^{-1}$, while for leftward perturbations, initial peaks ranged from −0.93s$^{-1}$ and −0.98s$^{-1}$. As the body responded, the rate of change of angular momentum reversed sign as participants attempted to apply counteracting torques and regain control, with peaks of up to −1.02s$^{-1}$ and −1.46s$^{-1}$ for rightward perturbations and peaks of up to + 0.98s$^{-1}$ and + 1.20s$^{-1}$ for leftward perturbations. Eventually, $\dot{H}$ approached zero as the participants recovered from the perturbation.

### Joint moment contributions

The moments exerted on the lower-limb joints of both legs are depicted in Fig 3 across the five experimental scenarios. In the *MI* and *MC* perturbation scenarios, the right hip peak adduction moment exhibited statistically significant changes of + 96% and −156.1% with respect to *CTRL*, respectively. These changes primarily occurred during the late stance of the right leg and the subsequent double support phase. Notably, in these perturbation cases, following the initial response from the right leg, the left leg's hip adduction also displayed significant changes

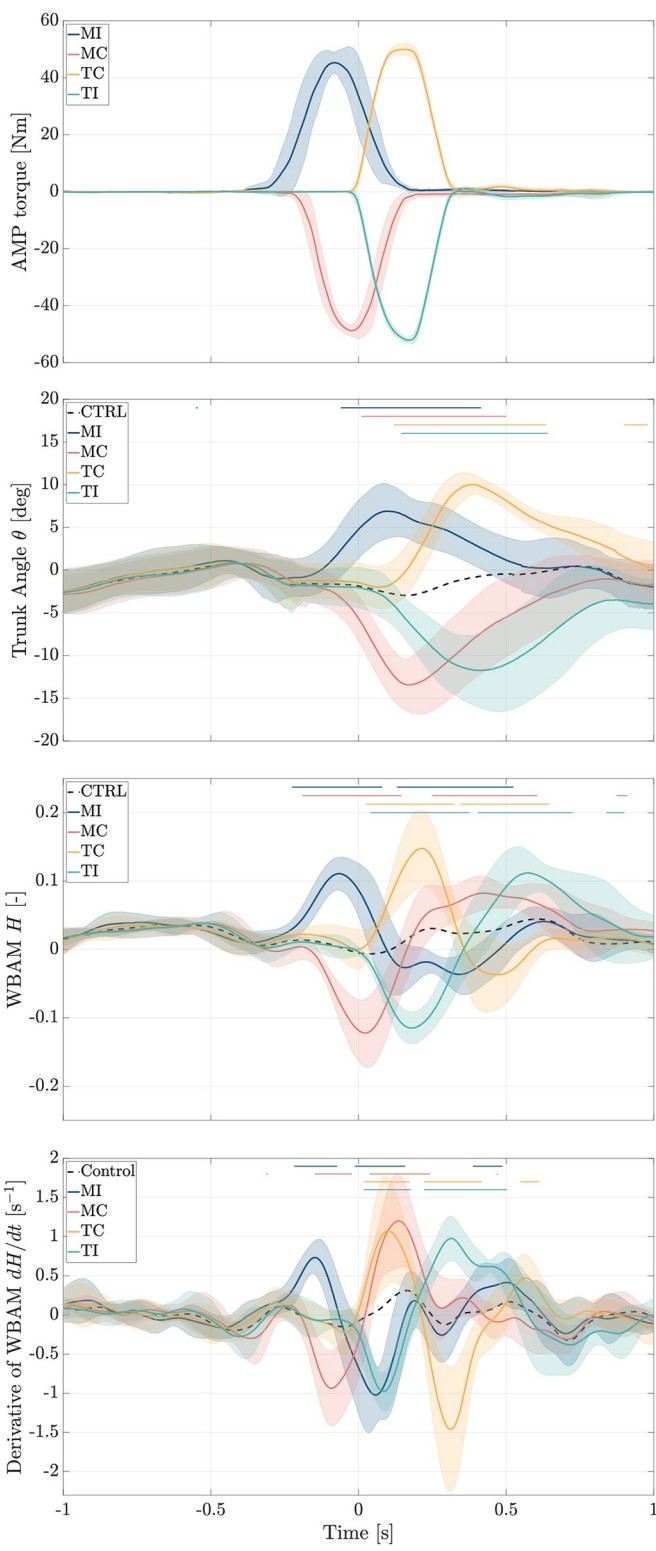

**Fig 2. Perturbation profiles and their effect.** Grand mean ± standard deviation of the perturbation profiles and their resulting effects on trunk angle (*θ*), whole-body angular momentum (WBAM, *H*), and time derivative of WBAM (*dH/dt*) in the frontal plane, averaged across all subjects and trials. In all figures, the time point 0 indicates the moment when the left leg comes into contact with the second force plate. Horizontal lines overlaid on the graphs indicate regions where signals in the perturbed cases exhibit significant differences compared to control trials (*p* < 0.05). *CTRL:*

Control Trials, *MI*: Midstance-Ipsilateral, *MC*: Midstance-Contralateral, *TC*: Touchdown-Contralateral, *TI*: Touchdown-Ipsilateral.

during the single support phase of the left leg. For *TC* and *TI* perturbation cases, left hip adduction demonstrated statistically significant changes of −105.4% and + 97.9%, respectively, occurring mainly during the left leg's single stance phase. Notably, in the *TI* case, the right leg's hip adduction also exhibited slight increases during its late swing phase. In addition to hip adduction, hip flexion showed statistically significant changes. Alterations were observed in the right leg flexion moment for *MI* and *MC*, and in the left leg hip flexion moment for *TC*. However, the magnitude of these changes is not particularly noteworthy. Furthermore, in the *MC* and *TC* cases, hip rotation of the left leg exhibited significant changes during the left single support phase.

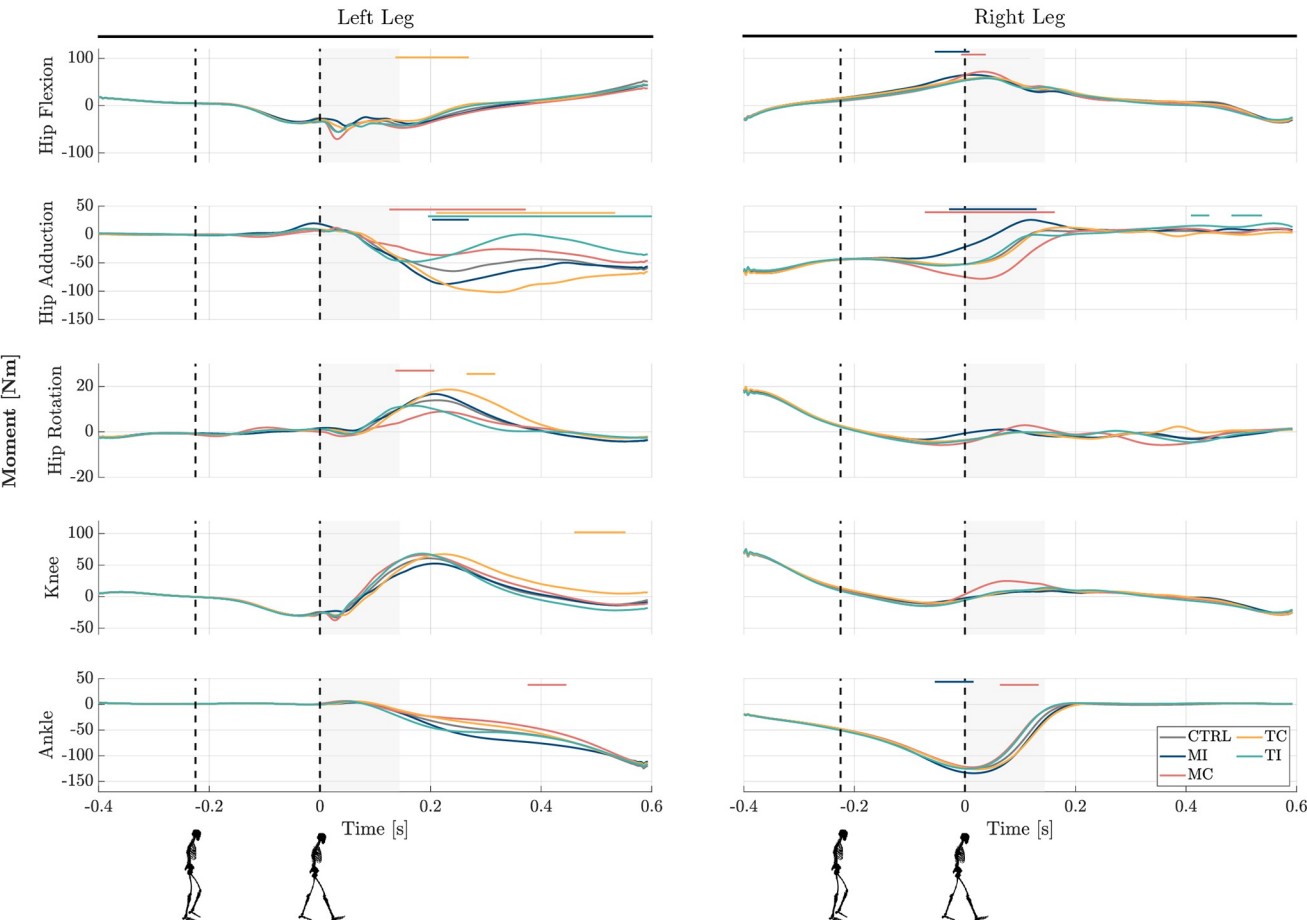

**Fig 3. Analysis of joints' reactive moments to mediolateral upper-body gyroscopic perturbations.** Each row displays the grand mean of moment time series for each joint (hip, knee, and ankle) for both the left and right legs across 10 subjects. Each figure depicts moments in five experimental conditions: Control (*CTRL*), Midstance-Ipsilateral (*MI*), Midstance-Contralateral (*MC*), Touchdown-Contralateral (*TC*), and Touchdown-Ipsilateral (*TI*). The time point 0 indicates the moment when the left leg comes into contact with the second force plate. Vertical dashed black lines mark the instance of perturbation occurrence. Horizontal lines overlaid on the graphs indicate regions where signals in the perturbed cases exhibit significant differences compared to control trials ($p < 0.05$). The shaded grey background depicts the estimated double support phase. The results are presented using coordinates following the convention established by OpenSim.

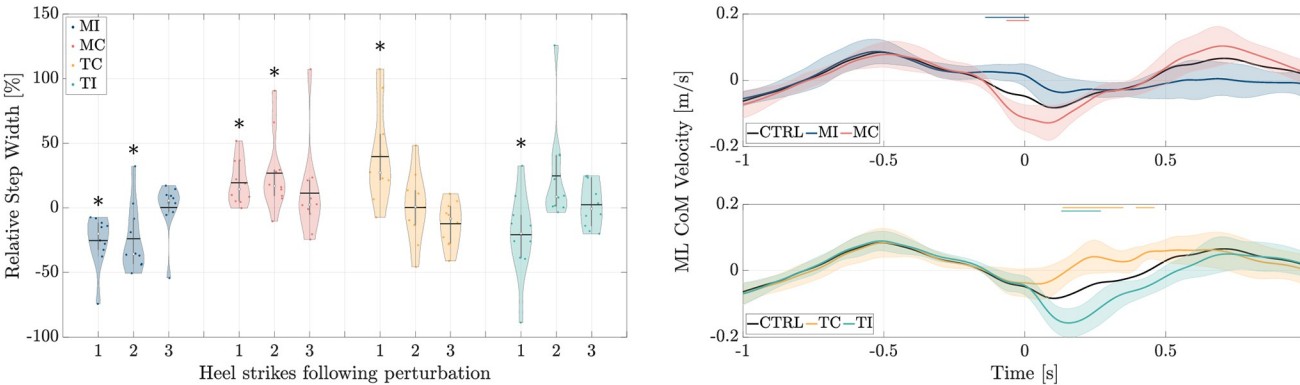

**Fig 4. Analysis of foot positioning.** (a) ML CoM velocity for all five experimental conditions; *CTRL*: Control Trials, *MI*: Midstance-Ipsilateral, *MC*: Midstance-Contralateral, *TC*: Touchdown-Contralateral, *TI*: Touchdown-Ipsilateral. The time point labeled as 0 indicates the moment when the left leg touches the second force plate. Horizontal lines overlaid on the graphs indicate regions where signals in the perturbed cases exhibit significant differences compared to control trials ($p < 0.05$). (b) step width at heel strike with respect to that of control trials for three steps following the perturbation step. Solid black lines and white circles in the violin plot represent the mean and median values, respectively. The asterisk symbol (*) denotes statistical significance at the $p < 0.05$ level with respect to the control trials.

In the knee joint, significant changes were observed only in the *TC* case for the left knee, and concerning the ankle, significant changes were noted in the *MI* and *MC* perturbation cases. No other notable differences were observed in the joint moments of the knee or ankle.

## Foot placement

Following the occurrence of ML perturbations, statistically significant changes in the CoM velocity are evident in the frontal plane (Fig 4a). Notably, for *MI* and *TC*, there was an increase in CoM velocity by 145.9% and 170.1%, respectively. Conversely, for *MC* and *TI*, a decrease in CoM velocity was observed, amounting to 136% and 166.1%, respectively.

Examining the impact on step width resulting from these perturbations, it is observed that for *MI* and *TI*, the step width decreases notably in the initial step post-perturbation, by 25.4% and 20.9% respectively (Fig 4b). In contrast, for *MC* and *TC*, there is an increase in step width in the subsequent step following perturbation, with increments of about 19.3% and 39.6% respectively (Fig 4b).

## Discussion

Insights into sensorimotor control of balance can be achieved by studying how individuals respond to controlled perturbations. Using an AMP to apply pure moments on the upper body and changing WBAM, we examined balance recovery schemes employed in coping with laterally perturbed human walking. The use of AMP allows for perturbing the WBAM with minimal effect on CoM excursions and WBLM changes. We found out that shortly after the perturbation onset, hip strategy and foot placement were the primary mechanisms for rejecting perturbation threats. These two mechanisms are the result of passive dynamics as well as controlled actions during the single and double support phases, which will be discussed in more detail in the following.

### Leading role of hip strategy

The analysis of lower-limb joint moments underscores the pivotal role played by hip abduction/adduction in generating reactive responses to mediolateral perturbations, particularly

during the single or double support phase. Notably, these reactive responses are dependent on the direction of the perturbation. In instances where the perturbation is directed towards the support stance leg (right leg for *MI* and left leg for *TI*), there is an increase in hip adduction to counteract the perturbation, whereas, when the perturbation is directed oppositely (*MC* and *TC*), hip abduction of the support leg increases to resist the perturbation; see Fig 3. These direction-specific adjustments in hip abduction/adduction moment highlight an active strategy employed by the body to counteract perturbations, aligning with previous research emphasizing the imperative role of active control in maintaining frontal plane balance [17]. Recognizing the significance of the hip's lateral degree of freedom in stabilization could lead to innovative designs of assistive devices, as recently explored [51–53].

Importantly, no significant responses were detected in other joints, underscoring the importance of the hip strategy in mitigating the effects of AMP perturbations. This observation aligns with findings from [40], which demonstrated a significant recovery initiated through the hip joint during stance following WBAM perturbation in the sagittal plane. Further, in standing experiments, previous works suggested a dominant role in regulating the hip joint moments in balance recovery from WBAM perturbations [9, 10]. More specifically, [9] showed that the hip and spinal moments provided the majority of the recovery response (up to nearly 85%). Our previous EMG analysis of the lower-limb muscles for the WBAM perturbations also highlighted that proximal muscles around the hip joint contribute the most to reject perturbation threats [42].

## Stepping strategy

In response to ML moment perturbations on the upper body, we observed that the ML CoM velocity at subsequent heel strike after perturbations correlates with the ML foot placement for all four perturbations cases. When the perturbation is applied towards the swing leg (*MC* and *TC*) it causes a change in the ML CoM velocity which corresponds to a wider step compared to the control trials. Conversely, for perturbations in the opposite direction (*MI* and *TI*), a shorter step width is observed; see Fig 4. This behavior is frequent for most subjects and is in line with previous works showing that the ML CoM velocity at heel strike has a direct relationship with ML foot placement [54–56].

Step width control is key to maintaining balance during gait, as ML foot placement is crucial due to limited CoP movement within the foot itself [19, 29]. Our analysis, depicted in Fig 3, reveals statistically significant changes in the hip adduction moment solely in the case of the *TI* condition during the swing phase, compared to the control conditions. Conversely, the remaining three perturbation scenarios show no statistically significant alterations in joint moments during the swing phase. This suggests that, despite experiencing lateral perturbation and active control of the stance leg, passive dynamics may suffice for foot placement. This finding aligns with prior research indicating that the dynamics of foot placement are likely influenced by both active control mechanisms and passive dynamics [56].

Furthermore, as depicted in Fig 4b, when perturbations occur at the mid-stance of the right leg (*MI* and *MC*), the step width is altered for the subsequent two heel strikes. In contrast, for perturbations at the touch-down of the left leg (*TC* and *TI*), significant changes in step width are only observed in the next heel strike after the perturbation. This distinction in the number of post-perturbation step adjustments can be attributed to the timing of perturbation. In *MI* and *MC* perturbations occurring shortly before the left leg makes contact with the ground, changing the width of the first step is an immediate response of the system's passive dynamics as there is no time for active modulation of the swing leg before the left leg touchdown (supported by EMGs [42]). However, more vulnerability to perturbation at single support (in *MI*

and *MC*) extends the influence in the next step by altering the step width, thereby leading to a two-step adjustment for recovery. The response is also observable in joint moment profiles (Fig 3) which shows that even after the left leg hits the ground, modulation of hip abduction/ adduction moment still persists based on the direction of the perturbation. On the other hand, in *TC* and *TI* cases where perturbations occur near double support, the body configuration is more resilient to perturbations. In such instances, a significant portion of the impact is dampened by the front limb, allowing for normal step width to resume after the first post-perturbation step.

The loss of balance during gait is sometimes accompanied by an altered stepping pattern [57]. However, in our case of AMP perturbations, no extra stepping was observed. This could be either due to the nature of moment perturbations (as opposed to force perturbations) or it is simply due to the fact that the magnitude of the perturbation was not high enough.

### Quick recovery after AMP perturbations

After inducing angular momentum perturbations in the upper body, our findings indicate that compensatory actions predominantly occur within a maximum of two steps post-perturbation. This observation aligns with our previous analysis of EMG activities [42]. Notably, the recovery from AMP perturbations is notably faster compared to other perturbation scenarios, such as platform translation, where a prolonged period of five to six steps was observed for complete recovery [57]. The accelerated recovery from AMP perturbations may be attributed to the human response dynamics to lateral perturbations, which are intricately linked to the instantaneous state of the body's momentum [58]. This suggests that humans prioritize recovery strategies [7], and recovery from angular momentum perturbations takes precedence over other types of perturbations [8].

### Methodological limitations

The AMP used in this study was initially designed to explore gyroscopic actuation principles and was therefore not optimized for weight or clinical applications. Consequently, its mass of 16kg could impose increased physiological strain on the trunk [59], potentially limiting its viability as an assistive device [60, 61]. Our previous analysis of the AMP's effect on nominal gait revealed that the device induces an average trunk flexion of 7.8deg compared to normal walking without the AMP [42]. This forward tilt of the trunk could influence hip and ankle moments [62]. Other changes in lower-limb joint kinematics were minimal, with deviations of less than 3deg compared to nominal gait [42]. Importantly, these alterations in gait do not affect the results of the current study, as the baseline (i.e., *CTRL*) for comparing different experimental conditions was established using the AMP device, rather than normal walking. In future work, to address the unwanted effects of the AMP's mass, we plan to use a newer version of the AMP weighing 4.9kg [63], which consists of two compact pint-sized CMGs, each weighing only 1.2kg [64].

### Conclusion

To conclude, our investigation reveals that able-bodied individuals exhibit a recovery pattern from WBAM perturbations primarily within the same stride as the perturbation occurred, leaving minimal compensatory actions for subsequent strides. Notably, this recovery is prominently driven by contributions from the hip joint, as indicated by alterations in the hip joint moment and appropriate mediolateral foot placement. These insights hold significant implications for the design and enhancement of controllers for robots and assistive devices, offering valuable considerations for clinical interventions.

Building on this work, our future studies will focus on two key areas. Firstly, we aim to improve the prediction accuracy of walking models based on the results observed in this study. Given that hip joint torque modulation and adjustments in the swing phase of the leg were identified as the primary compensatory responses to upper-body perturbations in our research, we propose incorporating control concepts such as Force Modulated Compliant Hip (FMCH) [65] for adjusting hip joint stiffness and Velocity-Based Leg Adjustment (VBLA) [66] for refining swing leg dynamics. Secondly, we will investigate how these improved models can provide better insights into human balance control. This could involve simulating various gait perturbations within the model and analyzing the resulting compensatory strategies. By comparing these model-predicted responses to actual human data, we can gain a deeper understanding of the underlying mechanisms of balance control.

## Acknowledgments

The authors would like to thank Andrew Berry and Christian Schumacher for their assistance in conducting the experiments and data acquisition.

## Author Contributions

**Conceptualization:** Omid Mohseni, Andre Seyfarth, Heike Vallery, Maziar A. Sharbafi.

**Data curation:** Omid Mohseni, Asghar Mahmoudi, Vahid Firouzi.

**Formal analysis:** Omid Mohseni, Maziar A. Sharbafi.

**Funding acquisition:** Andre Seyfarth, Heike Vallery.

**Investigation:** Omid Mohseni, Heike Vallery.

**Methodology:** Omid Mohseni, Andre Seyfarth, Heike Vallery.

**Project administration:** Andre Seyfarth, Heike Vallery, Maziar A. Sharbafi.

**Resources:** Andre Seyfarth, Heike Vallery.

**Software:** Omid Mohseni.

**Supervision:** Andre Seyfarth, Heike Vallery, Maziar A. Sharbafi.

**Validation:** Omid Mohseni.

**Visualization:** Omid Mohseni, Asghar Mahmoudi, Vahid Firouzi.

**Writing – original draft:** Omid Mohseni.

**Writing – review & editing:** Omid Mohseni, Asghar Mahmoudi, Vahid Firouzi, Maziar A. Sharbafi.

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
