## [Decision Letter · Decision Letter 0]

2 Oct 2024

PONE-D-24-24945Balance recovery schemes following mediolateral gyroscopic moment perturbations during walkingPLOS ONE

Dear Dr. Mohseni,

Thank you for submitting your manuscript to PLOS ONE. After careful consideration, we feel that it has merit but does not fully meet PLOS ONE’s publication criteria as it currently stands. Therefore, we invite you to submit a revised version of the manuscript that addresses the points raised during the review process.

 **Reviewers were overall positive about the manuscript, and although no significant technical issues have been recognized, I would suggest to revise the paper taking into account carefully the Reviewers comments and suggestions, since some concerns have been raised in particular regarding the added value of this work to existing literature, and the general impact.**

We look forward to receiving your revised manuscript.

Kind regards,

Alessandro Mengarelli

Academic Editor

PLOS ONE

**Journal Requirements:**

This work was supported by the German Research Foundation (www.dfg.de) within RTG 2761 LokoAssist under grant no. 450821862 and the Hessian Ministry of Higher Education, Science, Research and Art and its LOEWE research priority program under the grant ‘WhiteBox’, both awarded to AS. The funders had no role in the study design, data collection, data analysis and preparation of the manuscript.

3. Please note that your Data Availability Statement is currently missing the DOI/accession number of each dataset or a direct link to access each database. If your manuscript is accepted for publication, you will be asked to provide these details on a very short timeline. We therefore suggest that you provide this information now, though we will not hold up the peer review process if you are unable.

Reviewers' comments:

Reviewer's Responses to Questions

**Comments to the Author**

1. Is the manuscript technically sound, and do the data support the conclusions?

Reviewer #1: Yes

Reviewer #2: Yes

2. Has the statistical analysis been performed appropriately and rigorously? 

Reviewer #1: Yes

Reviewer #2: Yes

3. Have the authors made all data underlying the findings in their manuscript fully available?

Reviewer #1: Yes

Reviewer #2: Yes

4. Is the manuscript presented in an intelligible fashion and written in standard English?

Reviewer #1: Yes

Reviewer #2: Yes

5. Review Comments to the Author

**Reviewer #1:** This paper investigates how human test participants respond to pure WBAM perturbations in two directions and at two moments during the human gait cycle. The paper is very well written and nicely to the point.

The only feedback I have is rather minor:

L47: you contrast your mobile/portable AMP system perturbations with "traditional force-based and whole-body pitch angle perturbations". I understand your point of mobility versus statically connected to the treadmill(-base), now allowing you to do overground walking etc.. However, it is not really clear what you really mean with limitations of "force-based and whole-body pitch angle perturbations". Is it not possible to purely perturb AM with those systems? I think you could be a bit more explicit here.

Fig 1: I find Fig 1.c to be not so clear in what it is trying to show, since I cannot really see what foot is in the air or on the ground. Possibly both a front/side view might make it fully unambiguous. Another very minor remark: in the a) is a human, b) is a mannequin and c) is a skeleton. Possibly some consistency might be preferred (human silhouettes?).

Fig 2.:

- Maybe it is image compression imposed by the PLOS one reviewing system, but the figures are VERY compressed, almost to the extent that the text and lines in plots becomes very difficult to read (blurry) and distinguish: I cannot see/distinguish the H_CTRL and theta_CTRL lines.

- "across all subjects and trials", so we are looking at the mean? Please make a bit more explicit what we are really looking at. Why is the perturbation torque not trapezoidal (control bandwidth limitation? consequence of averaging?)

- If possible, also show (approximate) derivative of the WBAM, i.e. dH/dt, and perturbation torque together in 1 plot, (and possibly their estimated difference) because then we can infer the net ML CoM Torque that the participant generates to regulate the WBAM back to nominal behavior. Because afterwards, one can start thinking about how this feedback moment will be distributed over the joints, which will inform us more mathematically about relative weighting the participants solve this redundancy given the foothold constraints and future planning (leading to hip strategy).

Suggestion:

L241: a larger step width -> a wider step

L13,L56,L292 healthy -> able-bodied

Finally:

Do whatever you wish with the following piece of trivia (possibly ignore), but it could make Eq (1) shorter by 1 symbol. In Eq (1) there is no need to do "-v_body^COM". Those mass scaled contributions of the whole body com velocity all cancel out in the total sum. Consider 2 point masses (m1,m2) that have no inertia (tensor) that have their own com at r1, r2 and a barycenter (whole body com) at rc, with velocity vc (expressed in some frame). I use parameter A to add (A=1) or remove (A=0) the vc contribution. If one can show that the WBAM is independent of A, the point is made. Proof by MATLAB symbolic example:

syms m1 m2 A

r1 = sym('r1',[3 1]);r2 = sym('r2',[3 1]);

v1 = sym('v1',[3 1]);v2 = sym('v2',[3 1]);

rc = (r1*m1+r2*m2)/(m1+m2);vc = (v1*m1+v2*m2)/(m1+m2);

contribution1 = (cross(r1-rc,m1*(v1-A*vc)))

contribution2 = (cross(r2-rc,m2*(v2-A*vc)))

WBAM = simplify(contribution1+contribution2) %cancellations happen here

Which shows that WBAM is independent of A and vc can be omitted. This trivially generalizes to systems of N particles and also holds when product I*w is reintroduced for rigid bodies.

**Reviewer #2:** In this paper the authors reported on findings related to using a "backpack"-esq perturbation device which applies an angular momentum-based perturbation that can be used during overground walking. In this study the authors performed frontal plane perturbations.

In general the paper was well written. It appeared to be technically sound and rigorous.

In spite of being well written, I am left wondering about the "why" for this paper. On the one hand, I appreciate the effort to develop a perturbation device for overground walking. On the other hand, we already know a good deal about perturbation responses in healthy adults. The authors allude to implications for interventions for elderly individuals in their conclusion, however, I fail to see the direct relationship. We know from other populations that balance responses are not equivalent to healthy adults, so it would be reasonable to suspect that the strategies may not carry over to those elderly needing fall prevention interventions. So, I think the authors need to carefully reconsider the conclusions and implications of the present study after considering that other populations may not employ the same strategy as young healthy adults.

While no method of applying perturbations (currently) will result in unperturbed walking being 100% identical to overground walking, I would like to see some discussion about the degree to which walking with this device (unperturbed) correlates with regular, unperturbed overground walking. This is potentially important as 16kg is potentially a substantial load for some participants. In addition to the mass, it also appears rather "bulky" that may impact normal biomechanics. I could conceive that this would alter normal medio-lateral sway and / or increase the posterior position of the center of mass relative to a stepping foot. Moreover, this could easily result in alterations to steady-state walking momenta. Readers may have concerns about these effects. I understand that it appears that the authors have discussed this in their previous papers, but I, nevertheless, feel that a brief discussion is warranted in the present paper.

6. PLOS authors have the option to publish the peer review history of their article (what does this mean?). If published, this will include your full peer review and any attached files.

Reviewer #1: No

Reviewer #2: No

---

## [Author Response · Author response to Decision Letter 0]

25 Oct 2024

We would like to extend our sincere gratitude to the editor and the reviewers for their time and constructive feedback, which have greatly enhanced our manuscript. We have carefully considered all comments and made revisions to the paper accordingly. In response to the suggestions and concerns raised, we have modified the manuscript and included additional explanations and clarifications. For further details, please refer to the rebuttal letter submitted with the revised manuscript.

---

## [Decision Letter · Decision Letter 1]

26 Nov 2024

Balance recovery schemes following mediolateral gyroscopic moment perturbations during walking

PONE-D-24-24945R1

Dear Dr. Mohseni,

We’re pleased to inform you that your manuscript has been judged scientifically suitable for publication and will be formally accepted for publication once it meets all outstanding technical requirements.

Kind regards,

Alessandro Mengarelli

Academic Editor

PLOS ONE

Additional Editor Comments (optional):

Before sending the final version of the paper, please address the concern of one of the Reviewers about one of the figures.

Reviewers' comments:

Reviewer's Responses to Questions

**Comments to the Author**

1. If the authors have adequately addressed your comments raised in a previous round of review and you feel that this manuscript is now acceptable for publication, you may indicate that here to bypass the “Comments to the Author” section, enter your conflict of interest statement in the “Confidential to Editor” section, and submit your "Accept" recommendation.

Reviewer #1: All comments have been addressed

Reviewer #2: All comments have been addressed

2. Is the manuscript technically sound, and do the data support the conclusions?

Reviewer #1: Yes

Reviewer #2: Yes

3. Has the statistical analysis been performed appropriately and rigorously? 

Reviewer #1: Yes

Reviewer #2: Yes

4. Have the authors made all data underlying the findings in their manuscript fully available?

Reviewer #1: Yes

Reviewer #2: Yes

5. Is the manuscript presented in an intelligible fashion and written in standard English?

Reviewer #1: Yes

Reviewer #2: Yes

6. Review Comments to the Author

Reviewer #1: Authors have addressed all comments. In the Fig 2 bottom graph with dH/dt, the y-label is missing units (s^-1) (and the other plots seem to be missing vertical grid lines?). Good luck with finishing up the paper.

Reviewer #2: The authors have addressed all concerns I previously raised. I have no further concerns about this paper.

7. PLOS authors have the option to publish the peer review history of their article (what does this mean?). If published, this will include your full peer review and any attached files.

Reviewer #1: No

Reviewer #2: No

---

## [Editor Report · Acceptance letter]

16 Dec 2024

PONE-D-24-24945R1 

PLOS ONE

Dear Dr. Mohseni, 

I'm pleased to inform you that your manuscript has been deemed suitable for publication in PLOS ONE. Congratulations! Your manuscript is now being handed over to our production team.

Kind regards, 

on behalf of

Dr. Alessandro Mengarelli 

Academic Editor

PLOS ONE